# Identifying interventions to reduce peripartum haemorrhage associated with caesarean delivery in Africa: A Delphi consensus study

**APORG Caesarean Delivery Haemorrhage Group**[¶]*

¶ Membership of APORG Caesarean Delivery Haemorrhage Group is provided in the Acknowledgments and S1 Text.

* bruce.biccard@uct.ac.za

**Data Availability Statement:** Data are provided as part of the Supporting Information.

## Abstract

Women in Africa are fifty times more likely than in high-income settings to die following caesarean delivery, and peripartum haemorrhage is most strongly associated with mortality. We aimed to establish consensus on which interventions are considered most feasible to implement and most effective at reducing haemorrhage associated with caesarean delivery across Africa. We conducted a Delphi consensus study, including obstetric and anaesthesia providers from across Africa. In round one the expert group proposed key interventions for consideration. In rounds two and three the interventions were ranked on a 9-point Likert scale for effectiveness and feasibility. Round four was an online discussion to establish consensus on effectiveness and feasibility of interventions for which this had not been reached in round three. Twenty-eight interventions were considered both highly effective and feasible in Africa. Interventions covered a range of fields, categorised into direct- or indirect interventions. Direct interventions included: risk assessment and screening; checklists and protocols; monitoring and surveillance; availability of resources; ability to perform technical skills. Indirect interventions included: community and maternal education; contraception and family planning; minimum training standards; referral patterns and delays; advocacy to key stakeholders; simulation and team training; and 24-hour access to safe emergency caesarean delivery. Interventions considered both effective and feasible in reducing peripartum haemorrhage associated with caesarean delivery in Africa were identified. A multi-layered implementation strategy, including immediately developing a perioperative caesarean delivery bundle of care, in addition to longer-term public health measures may have a profound impact on maternal mortality in Africa.

**Funding:** SM acknowledges the South African Medical Research Council Mid-career Scientist Award. EHT acknowledges the Africa Oxford Initiative (AfiOx-188). The funders had no role in study design, data collection, data analysis, decision to publish, or preparation of the manuscript.

**Competing interests:** The authors have declared that no competing interests exist.

## Introduction

In sub-Saharan Africa one in 36 women die during childbirth—five times the global average [1]. This equates to 545 maternal deaths per 100,000 live births–accounting for two thirds of all maternal deaths globally [1], which is substantially higher than the Sustainable Development Goal for 2030 of <70 per 100,000 live births [2]. Caesarean delivery rates below 19% of live births are associated with increased maternal mortality [3]. In Africa it is estimated that 7.3% of live births are by caesarean delivery, while in sub-Saharan Africa the rate is as low as 3.5% [4].

Despite the low caesarean delivery rate, the African Surgical Outcomes Study (ASOS) found that caesarean delivery is the most common surgical procedure, accounting for one third of surgical procedures performed [5]. Women undergoing caesarean delivery in Africa were 50 times more likely to die compared to women in high-income settings. Haemorrhage accounted for 70% of perioperative complications and was independently associated with mortality [6].

The African Perioperative Research Group (APORG) highlighted the *'early identification and management of mothers at risk from peripartum haemorrhage in the peri-operative period'* as a continental research priority [7]. Interventions to reduce maternal mortality exist [8], although it is unclear how implementable they are in low resource environments. To respond to the need to improve maternal outcomes in Africa, we aimed to establish consensus on which interventions are recommended by African clinicians and researchers as the most feasible to implement and most effective at reducing peripartum haemorrhage associated with caesarean delivery.

## Materials and methods

The Delphi technique is a widely accepted method for generating data and establishing consensus among subject matter experts. The objective of this study was to use the Delphi process to reach consensus on interventions to decrease haemorrhage during and after caesarean delivery, which are considered effective and feasible to implement across Africa.

### Ethics and consent

Ethics approval was provided by the Human Research Ethics Committee of the Faculty of Health Sciences of the University of Cape Town, South Africa (HREC 187/2020), and by the Oxford Tropical Research Ethics Committee (524–20). Collaborators were provided with an information sheet and online consent was obtained at the beginning of each round.

### Data collection

The study was conducted in English and French. Study data from rounds one to three were collected and managed using REDCap, a web-based data capture software application [9]. Round four was held via ZOOM Video Communications Version: 5.0.5, an online video conferencing platform.

### Participant selection

To identify African obstetricians and anaesthetists considered leaders in the field of obstetric care and obstetric anaesthesia, we asked National leaders from previous APORG studies—the African Surgical Outcomes Study [5] and/or the African Surgical OutcomeS-2 Trial [10] to identify and invite experts from within their respective countries. We encouraged the participation of two obstetric and anaesthesia providers from each country.

## Delphi process

**Round one.** In round one, participants were asked to propose at least six key interventions for consideration. Participants were asked to identify interventions across three broad categories: prevention, early identification, and management of peripartum haemorrhage in the perioperative period. Participants were encouraged to provide justifications and references for the inclusion of recommended interventions. The list of interventions was amalgamated and grouped according to common themes by ET, BB and SM. We collected demographic data on participants' speciality and their healthcare facility level.

**Round two.** In round two, all recommended interventions were distributed to participants. Participants were asked to score each intervention on a 9-point Likert scale for (i) effectiveness–'*how effective do you predict the intervention would be at reducing peripartum haemorrhage in the perioperative period, within your local context*', and (ii) feasibility–'*how feasible do you predict it would be to implement the intervention within your local context*'. A score of 1–3 was considered low effectiveness/feasibility (not recommended); 4–6 was considered moderate effectiveness/feasibility (possibly recommended); and 7–9 was considered high effectiveness/feasibility (recommended). Participants were encouraged to provide further justifications or references to support their scoring. The group scores for each intervention were summarised by the median and interquartile range (IQR).

**Round three.** In round three, participants were presented with the same list of recommended interventions, alongside summarised group scores (median and IQR) and anonymised justifications or references from round two. Participants were asked to re-score each intervention on the same 9-point Likert scale for effectiveness and feasibility, while considering the summarised group scores from the previous round. Participants were again given the option to provide justifications and/or references for their score. The group scores for each intervention were again summarised by the median and IQR.

**Round four.** Round four was an online discussion. Interventions were presented to participants based on the summarised group scores from round three. Interventions with a median score of 1–3 in either effectiveness or feasibility domain were excluded and not considered in round four. Interventions with a median score of 7–9 in both domains were recommended and included in the final list of interventions. The aim of round four was to gain consensus on whether to upgrade any of the 'potential interventions' with a median score of 4–6 in one or more domain to the final list of recommended interventions. The study process is summarised in Fig 1.

## Results

The Delphi process was conducted from August to November 2020. Overall, 55 experts from 24 African countries participated: Angola; Botswana; Burkina Faso; Burundi; Democratic Republic of Congo; Egypt; Ethiopia; Gambia; Ghana; Kenya; Libya; Madagascar; Malawi; Mali; Mauritius; Mozambique; Niger; Nigeria; Rwanda; Sierra Leone; South Africa; Tanzania; Uganda; Zambia (Fig 2).

In round one, 44 experts participated. 20/44 (45%) were obstetric providers and 24/44 (55%) were anaesthesia providers. 34/44 (77%) of participants worked in a tertiary facility, 8/44 (18%) at a regional facility and 2/44 (5%) at a district/rural facility. Participants recommended 195 potential interventions. The interventions were de-duplicated and amalgamated to form 39 unique interventions across 4 broad categories of interventions: community based; antenatal care and assessment; perioperative care, and health system-strengthening interventions. In round two, 41 participants scored each intervention on the 9-point Likert scale for effectiveness and feasibility. In round three, 40 participants re-scored the interventions, while considering

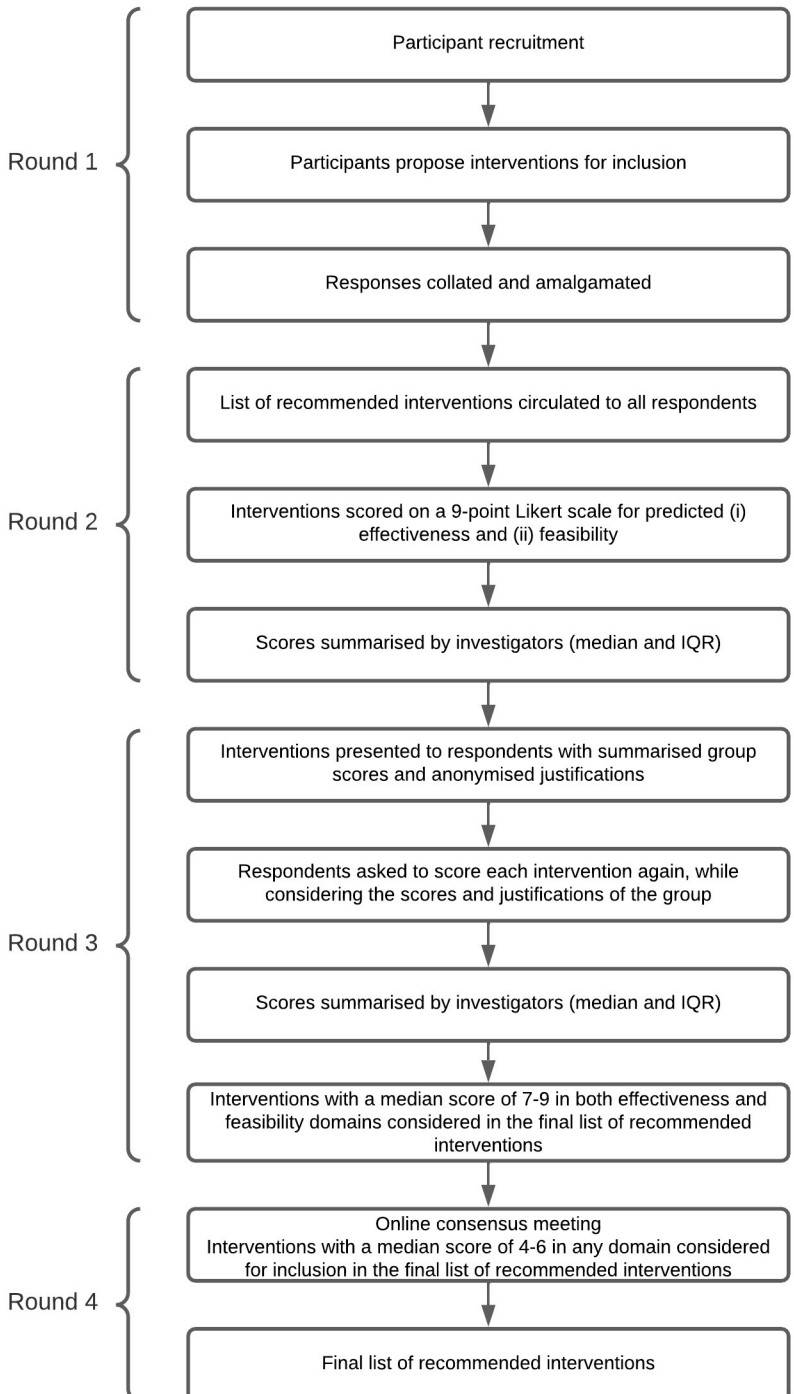

**Fig 1. Study process flow diagram.**

the group scores (median and IQR) (S1 Data) and anonymised justifications/references from round two. Following round three, 24 interventions were of high effectiveness (median 7–9) and high feasibility (median 7–9) and were recommended in the list of effective and feasible interventions. One intervention was of low feasibility (median 1–3) and was not considered for the final list of recommended interventions. There were 14 potential interventions considered

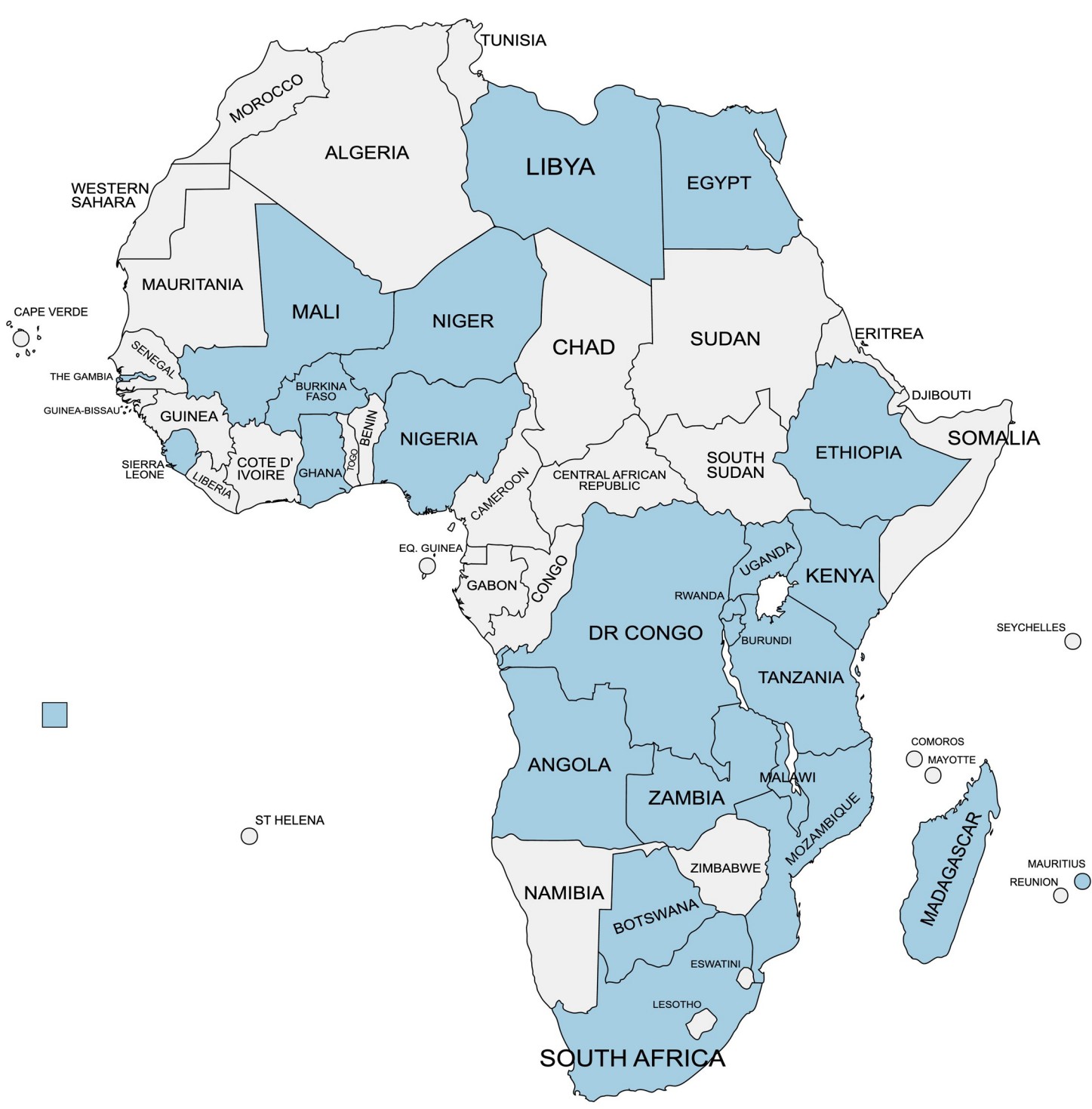

Created with mapchart.net

**Fig 2. Map of countries represented (https://www.mapchart.net/).**

to have moderate effectiveness (median 4–6) and/or moderate feasibility (median 4–6), for consideration in the round four consensus meeting. In round four, 30 participants took part in the online discussion. Four 'potential interventions' were upgraded to the final list of recommended interventions. The summarised group scores for each intervention are listed in S1 Data. The final recommendations include 28 interventions, listed in Table 1. These interventions were categorised into either direct- or indirect interventions. Direct interventions included: pre-operative risk assessment and screening; checklists and protocols; monitoring and surveillance; availability of resources; ability to perform technical skills. Indirect interventions included: community and maternal education; availability of contraception and family planning, minimum

**Table 1. Recommended interventions.**

| Direct interventions |
|---|
| Antenatal care and assessment |
| Routine preoperative risk assessment and risk stratification for peripartum haemorrhage |
| Routine screening for anaemia and patient blood management programme |
| Routine antenatal assessment by the anaesthesia provider for all women for planned caesarean delivery |
| Maternal education on the early self-recognition of signs and symptoms of peripartum haemorrhage |
| Perioperative care |
| Routine use of a Surgical Safety Checklist modified for use at caesarean delivery |
| Implementation of a standardised peripartum haemorrhage protocol |
| Intrapartum monitoring: routine, close monitoring of mother and fetus during intrapartum period |
| Early and increased surveillance of vital signs in the postpartum period, possibly in a dedicated environment |
| Routine use of modified early obstetric warning score in the postoperative period, or routine active monitoring for haemorrhage e.g. shock index |
| Early involvement of anaesthesia providers in the management of patients at risk. |
| Availability of a peripartum haemorrhage 'package' containing all items required to manage postpartum haemorrhage |
| Availability of emergency blood products |
| Availability of first line uterotonic agents |
| Availability of alternative/second line uterotonic agents |
| Availability of tranexamic acid |
| Ability to perform active management of the third stage of labour |
| Ability to perform uterine massage and bimanual compression |
| Ability to perform hysterectomy and uterine artery ligation |
| Ability to perform B-lynch- and compression sutures |
| Ability to perform balloon tamponade |
| Indirect interventions |
| Community based |
| Community information, education and communication regarding reproductive health (e.g., knowledge on risks, the early recognition of danger signs in pregnancy, and the advantages of seeking healthcare at an early stage) |
| Availability of contraception and family planning |
| Health system-strengthening |
| Ensuring a minimum training standard for healthcare providers |
| Advocating to key stakeholders about the importance of reducing maternal morbidity and mortality related to peripartum haemorrhage |
| Establishing smooth and timely referral patterns between levels of care |
| Reducing delays in receiving care once at the healthcare facility |
| Multidisciplinary simulation and team training |
| Availability of resources to perform caesarean delivery 24 hours a day |

training standards; referral patterns and delays; advocacy to key stakeholders; simulation and team training; and 24-hour access to caesarean delivery.

## Discussion

We used a Delphi process to identify effective and feasible interventions to reduce haemorrhage during and after caesarean delivery in Africa. Twenty-eight interventions were recommended and considered both highly effective and feasible. The expert group provided representation from the disciplines of obstetric and anaesthesia in 24 African countries. Interventions were divided into (i) direct interventions, including: antenatal care and assessment, and perioperative care components; and (ii) indirect interventions, including: community based and health systems-strengthening components. The recommendations were multi- or interdisciplinary; in addition to clinician-focussed interventions, many public health-related interventions were recommended–suggesting a holistic systems-strengthening approach.

### Direct interventions—Antenatal care and assessment

Anaemia during pregnancy is associated with an increased risk of post-partum haemorrhage; the risk of maternal mortality in women with severe anaemia has an odds of 2.36 when compared to women without the condition [11]. Early identification of anaemia through screening would facilitate blood management programmes.

Routine preoperative risk assessment and risk stratification for peripartum haemorrhage may facilitate preoperative optimisation and/or the implementation of strategies to mitigate risks in the perioperative period. Current guidance on haemorrhage risk stratification is defined by the Royal College of Obstetricians and Gynaecologists, and may not be applicable in its entirety in low resource settings [12].

Midwives play a critical role in providing antenatal care, including the assessment of risk factors. A modelling study projected a 39% reduction in maternal mortality in low human development index (HDI) countries, and a 43% reduction in maternal mortality in low-to-medium HDI countries by 2035 if midwife-delivered interventions were to be scaled up by a 'substantial amount', defined as a 25% increase in coverage every five years [13].

Comprehensive antenatal care presents an opportunity to provide maternal education on the self-recognition of signs and symptoms of peripartum haemorrhage. A study in Burkina Faso, Ghana and Tanzania showed that a third of women were not counselled on the danger signs of a high-risk pregnancy, and the proportion of women retaining the counselled information was generally low [14]. Context-sensitive and tailored methods of relaying information on the self-recognition of danger signs may reduce the 'phase one delay' in deciding to seek healthcare [15].

### Direct interventions—Perioperative care

Use of surgical safety checklists have been associated with decreased postoperative mortality and decreased surgical complication rates. A meta-analysis of randomised trials assessing the effect of a surgical safety checklist found a statistically significant reduction in postoperative mortality, yet none of the included studies were from Africa [16]. Despite the simplistic nature of checklists, there are inherent challenges to implementation within complex health systems. An analysis of pooled data from five prospective cohort studies indicated significant variation in uptake of the World Health Organisation (WHO) Surgical Safety Checklist (SSC) across the world. Surgical patients in the African Surgical Outcomes Study were exposed to the SSC in 57.1% of operations [5], below the pooled global average of 75.4% [17]. The WHO SSC was significantly less likely to be used in obstetric and gynaecological surgery compared to abdominal

surgery in the pooled global analysis [17]. Given that use of the WHO SSC is low in Africa and in obstetrics, modified SSCs that are tailored to incorporate factors specific to caesarean delivery and practice in Africa, with context-sensitive implementation strategies, may have better uptake and utilisation. A study in a rural Rwandan hospital implemented a modified WHO SSC, adapted for low resourced settings [18]. Following implementation of the checklist, there was an increase in the documentation of estimated blood loss, reduced hospital length of stay, and an increase in the administration of antibiotics prior to incision. During the study period, the checklist was used in over 80% of procedures [18]. This indicates the importance of developing context-sensitive implementation strategies alongside checklists tailored for differing surgical specialties and environments.

ASOS showed that the high mortality associated with caesarean delivery across Africa is likely associated with 'failure to rescue' [6]–defined as the hospital death rate following a postoperative adverse outcome or complication. Women in Africa are three times more likely to experience a complication following caesarean delivery, yet are 50 times more likely to die than women in high-income settings. This indicates a failure to rescue rate 17 times higher in Africa than in high-income settings [6]. The implementation of interventions which improve a health system's 'capacity to rescue', may reduce the failure to rescue associated with caesarean delivery.

Delphi experts identified interventions which form a framework to improve capacity to rescue (Fig 3). The framework divides interventions into two groups: (i) early detection of the complication, and (ii) timely and appropriate management.

For 'early detection of the complication', the expert group recommend both 'early and increased surveillance of vital signs in the perioperative period, possibly in a dedicated

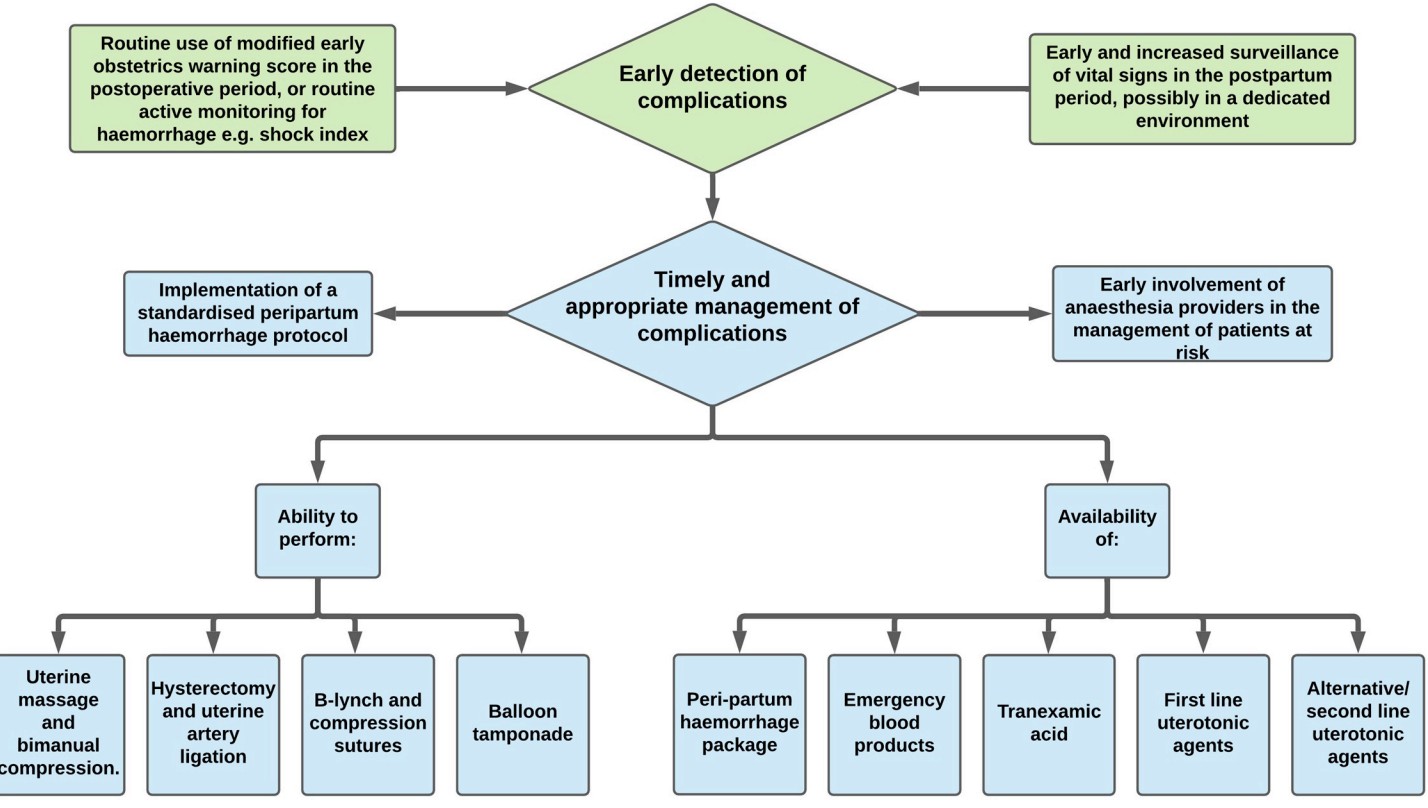

**Fig 3. Improving the capacity to rescue intervention framework.**

environment' and 'routine use of modified early obstetric warning score in the postoperative period, or routine active monitoring for haemorrhage e.g., shock index'. Given the challenges in estimating volume of blood loss, and that blood loss is often underestimated on visual assessment, vital signs are a critical indicator of haemodynamic instability [19]. The group highlighted the importance of validated objective scoring systems, enabling prompt and appropriate escalation of care. It is important that scoring systems are applicable to patient cohorts in Africa, and are feasible for use in low-resource settings. Due to the normal physiological changes of pregnancy, which include an increased blood volume, women may experience a prolonged period of apparent physiological compensation, prior to rapid deterioration [20]. General scoring systems may therefore be inappropriate in the obstetric population, and use of an obstetric-specific modified early obstetric warning score may be more effective. Modified early obstetric warning scores have been developed for- and embedded within clinical practice in high-income settings [20]. There is a need to develop a modified early obstetric warning score specific for use in a low-resource setting, and validated on local caesarean delivery populations. The shock index is a rudimentary tool based upon the measurement of systolic blood pressure and heart rate. Despite its simplicity, shock index is a good predictor of blood transfusion and admission to an intensive care unit following haemorrhage [19]. The Delphi group highlighted the importance of 'routine' use, which emphasises the need to embed such interventions within patient safety culture, as opposed to intermittent use at the discretion of the healthcare provider. According to the African Surgical Outcomes Study, inadequate human and physical resources are key factors in the increased rate of failure to rescue across Africa. User-friendly objective surveillance scores may be utilised by a range of healthcare personnel, and not limited to use by trained doctors and nurses. Given the sparse workforce density of surgical, anaesthesia, nursing and midwifery providers across Africa, such tools may be impactful.

The second component of the failure to rescue interventions framework focusses on timely and appropriate management of complications. Recommendations refer to the ability to perform clinical interventions such as: uterine massage and bimanual compression; hysterectomy and uterine artery ligation; B-lynch and compression sutures; and balloon tamponade. These are life-saving procedures which require trained personnel and physical resources. Uterotonic agents and tranexamic acid have been proven to be effective for the treatment of postpartum haemorrhage, though supply may vary in many low-resource settings [21]. Emergency blood products are a scarce resource across Africa. A review estimated that 26% of obstetric haemorrhage-related death in Africa was due to lack of available blood transfusion [22]. Rapid access to emergency blood transfusion is life-saving, but relies on a complex health system for its procurement. The establishment of a peripartum haemorrhage package implies the availability of necessary emergency resources, and the logistical ease of timely access. The use of emergency packages/bags/trolleys for time-critical situations is well established in resuscitation scenarios. The package may also contain protocols. The implementation of checklist-based haemorrhage protocols are associated with a reduction in maternal morbidity [23]. Obstetric haemorrhage packages/bundles exist [24, 25] though the development and assessment of packages/bundles contextualised for caesarean delivery in low-resource settings is needed. The 'early involvement of anaesthesia providers in the management of patients at risk' acknowledges the importance of the early escalation of care and the vital role of anaesthesia providers in the management of haemodynamically unstable patients.

The range of recommended interventions within the failure to rescue framework may disrupt the cascade of events that occur from complication to death, and increase the capacity to rescue following caesarean delivery in Africa.

## Indirect interventions—Community based

The expert group recommended 'Community information, education and communication regarding reproductive health'. Multiple community factors and perceptions regarding obstetric complications have been identified as barriers to utilising emergency obstetric care in Africa. Community barriers to accessing obstetric care identified in the literature include: socio-cultural factors, confidence in alternative practices, negative views of healthcare services, distrust in health care workers, stigma regarding seeking healthcare, anticipating self-improvement without seeking healthcare, fear of caesarean delivery and blood transfusion, and preference for delivery in the home environment [26]. Deciding to seek care may be impacted by perceptions of the pregnant woman, their family and the wider community, and thus information, education and communication-based interventions aimed at a wider community level may reduce the 'phase one delay', in which time is spent deciding whether to seek care [15].

Both community information and education, and access to contraception and family planning, form critical components of the Sustainable Development Goal 3.7: *"Target 3.7: By 2030, ensure universal access to sexual and reproductive health-care services, including for family planning, information and education, and the integration of reproductive health into national strategies and programmes"* [27]. Modern contraceptives form a vital primary prevention strategy to reduce maternal mortality by averting maternal mortality directly (by decreasing the total fertility rate and the absolute number of maternal deaths) and indirectly (by reducing the frequency of high-risk, high-parity pregnancies) [28]. Importantly, Africa has the lowest prevalence of contraceptive use, and the highest unmet need for family planning in the world [29]. A universal, community based approach to improve access to and uptake of modern contraceptives and family planning services may therefore be particularly effective.

## Indirect interventions—Health system-strengthening

Multidisciplinary simulation and team training designed to optimise team performance, may improve team morale, communication and utilisation of patient safety protocols. Context-sensitive Non-Technical Skills for Surgeons (NOTSS) training, with emphasis on specific challenges faced within various resource settings, have been developed and feasibility-tested in Rwanda [30].

Delays have been well described as a contributing factor to maternal mortality [15]. This Delphi process recommended a focus on interventions which reduce delays in receiving care once at the healthcare facility, and establishing smooth and timely referral systems between levels of care. This may reflect the make-up of the Delphi group, which recommended addressing delays which can be more directly influenced by obstetric and anaesthesia providers. A meta-analysis of maternal mortality after caesarean delivery found that women are more likely to die at teaching and tertiary hospitals [31]. National audit data from the National Committee on Confidential Enquiries into Maternal Deaths in South Africa show that many caesarean deliveries were performed at district hospitals and when complications arose women were subsequently referred to higher levels of care (regional, tertiary and national hospitals), where they then died [32]. These data emphasise the need to improve referral systems, to ensure timely referral and transfer of women at high risk of peripartum haemorrhage. There may be a role for technological interventions to improve communication between healthcare workers at different levels of care, an example being the model adopted by SURG-Africa—implementing WhatsApp group support networks to guide appropriate referrals between levels of care [33].

Emergency caesarean delivery may be required at any time of day. The ability to provide a safe 24-hour caesarean delivery service may reduce delays in receiving care and unsafe referrals between healthcare facilities. In addition, improving 24-hour access to caesarean delivery may

contribute to increasing the caesarean delivery rate across Africa, to rates associated with a reduction in maternal mortality [3, 4].

Minimum training standards should be enforced for healthcare providers. A lack of appropriately trained doctors and nurses continues to be the most frequently cited avoidable risk factor for maternal mortality in South Africa [32].

The expert group identified the importance of 'advocating to key stakeholders about the importance of reducing maternal morbidity and mortality related to peripartum haemorrhage'. Many of the recommended interventions are varied, inter-disciplinary and reflect a system strengthening approach, in addition to a clinician-focussed approach. Strengthening of healthcare systems is complex and requires buy-in from key stakeholders, with substantial upstream support at government and policy-maker level. Frameworks which measure the impact of such interventions within the wider healthcare system, may be of use [34].

The rising number of caesarean deliveries across Africa, combined with the unacceptably high proportion of women who die following this procedure [4, 6] necessitates urgency when advocating for interventions to strengthen health systems, to prevent avoidable deaths of mothers across the continent.

## Strengths

This was an African continental consensus study representing 24 countries, with a high response rate, and with representation from anaesthesia and obstetric providers. Rounds one to three were conducted in French and English to increase accessibility and participation.

## Limitations

No patients, midwives or public health specialists were involved in the Delphi process. Despite this, the suggested interventions cover broad health domains. The involvement of these stakeholders may have widened the scope of recommendations.

Since the Delphi study was conducted online, this study is biased towards receiving data from those with access to email and the internet, and the majority of input was received from participants working in tertiary healthcare settings. This may have resulted in the omission of critical insights from those working in district and rural healthcare facilities.

## Conclusion

This Delphi study recommends interventions considered both effective and feasible in reducing haemorrhage during and after caesarean delivery in Africa. A multi-layered implementation strategy, which includes immediately developing a perioperative caesarean delivery bundle of care, in addition to the implementation of longer-term public health measures may have a profound impact on maternal mortality in this under-resourced continent.

## Supporting information

**S1 Text. Members of the APORG Caesarean Delivery Haemorrhage Group.**
(DOCX)

**S1 Data. Summarised Delphi scores.**
(DOCX)

## Acknowledgments

Members of the APORG Caesarean Delivery Haemorrhage Group:

Elliott H Taylor, Salome Maswime, David Bishop, Fred Bulamba, Rowan Duys, Robert Dyer, Sue Fawcus, Thomas O Konney, Milton W Musaba, Jolene Moore, Dolly M Munlemvo, Akinyinka Omigbodun, Kokila Lakhoo, Bruce M Biccard, Freddy F Kabambi, Dawid van Straaten, Leyandis Cobas, Tadele Melese Benti, Mamo Woldu Kassa, Gaone Kediegile, Ouedraogo Nazinigouba, Ndayisaba Carter, Harerimana Salvator, Mubeya Franck, Ted B Likongo, Mukenga Mamba Martin, Maher Fawzy, Ashraf Nabhan, Melat Sebsibie, Rediet Shimeles Workneh, Alex N Bosire, Timothy M Mwiti, Rabie Salem Alfetouri, Rajaonarison Tahina Joëlle, Razafindrainibe Tanjonirina, Delia C Mabedi, Priscilla Mvula-Mtila, Boubacar Diallo, Moustapha Issa Manganè, Vakil Leellodharry, Emila Jeque, Magda Ribeiro, Nayama Madi, Idrissa Rekia, Abiodun Aboyej, Magnifique Irakoze, David Ntirushwa, Mvukiyehe Jean Paul, Eugene Tuyishime, Valerie John-Cole, Omobowale G. Olopade, Ahmadu Sesay, Motselisi Mbeki, Felicia Molokoane, Gerald Cubwa, Gloria Kinasa, Letisia Frostan Komba, Amos Mazuka, Sulayman Jallow, Abdoulie Keita, Musa Marena, Anna Njie, Masirending Njie, Ushmaben Patel-Mujajati, Angel Phiri *(please refer to S1 Text. Members of the APORG Caesarean Delivery Haemorrhage Group—for author affiliations).*

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
