## [Decision Letter · Decision Letter 0]

9 Mar 2022

PGPH-D-22-00210

Identifying interventions to reduce peripartum haemorrhage associated with caesarean delivery in Africa: a Delphi consensus study

Dear Dr. Biccard,

Thank you for submitting your manuscript to PLOS Global Public Health. There are a few relatively minor points brought up by the reviewers that I think should be fairly easy to address. Therefore, we invite you to submit a revised version of the manuscript that addresses these points.

We look forward to receiving your revised manuscript.

Kind regards,

M. Dylan Bould

Academic Editor

Journal Requirements:

1. Please provide separate figure files in .tif or .eps format only.  Please ensure that all files are under our size limit of 20MB.  

For more information about how to convert your figure files please see our guidelines: Once you've converted your files to .tif or .eps, please also make sure that your figures meet our format requirements

2. Please provide us with a direct link to the base layer of the map used in Fig 2 and ensure this location is also included in the figure legend. 

Please note that, because all PLOS articles are published under a CC BY license (creativecommons.org/licenses/by/4.0/), we cannot publish proprietary maps such as Google Maps, Mapquest or other copyrighted maps. If your map was obtained from a copyrighted source please amend the figure so that the base map used is from an openly available source.

Please note that only the following CC BY licences are compatible with PLOS licence: CC BY 4.0, CC BY 2.0  and CC BY 3.0, meanwhile such licences as CC BY-ND 3.0 and others are not compatible due to additional restrictions. If you are unsure whether you can use a map or not, please do reach out and we will be able to help you. 

The following websites are good examples of where you can source open access or public domain maps:

Additional Editor Comments (if provided):

Reviewers' comments:

Reviewer's Responses to Questions

**Comments to the Author**

1. Does this manuscript meet PLOS Global Public Health’s publication criteria? Is the manuscript technically sound, and do the data support the conclusions? The manuscript must describe methodologically and ethically rigorous research with conclusions that are appropriately drawn based on the data presented.

Reviewer #1: Yes

Reviewer #2: Yes

Reviewer #3: Yes

2. Has the statistical analysis been performed appropriately and rigorously?

Reviewer #1: I don't know

Reviewer #2: Yes

Reviewer #3: Yes

3. Have the authors made all data underlying the findings in their manuscript fully available (please refer to the Data Availability Statement at the start of the manuscript PDF file)?

Reviewer #1: Yes

Reviewer #2: Yes

Reviewer #3: Yes

4. Is the manuscript presented in an intelligible fashion and written in standard English?

Reviewer #1: Yes

Reviewer #2: Yes

Reviewer #3: Yes

5. Review Comments to the Author

Reviewer #1: Thank you for the opportunity to review this manuscript of a Delphi consensus to establish a context-specific list of feasible and effective strategies to reduce mortality from peripartum hemorrhage with caesarean section in Africa. This research is timely and necessary within the Sustainable Development Goals framework and the African Perioperative Research Group priorities. The variety of interventions considered, ranging from the provider to systems level, was highly pragmatic and comprehensive. It was informative to read about specific examples of recommendations that have been successfully implemented and sustained in various African countries. The paper also serves as a robust foundation for future work focused on operationalizing these interventions.

I have some comments to be considered:

1. Background: The authors state that it is unclear whether current interventions to reduce maternal mortality are implementable in low resource settings. The authors could provide more information about pre-existing guidelines with focus on who developed them and the intended target population, to strengthen the argument that they may not be applicable to their specific context in Africa.

2. Methods: The non-random sampling technique should be further described. Why did the authors decide to only approach leaders from the African Surgical Outcomes Study and/or African Surgical Outcomes-2 Trial? Were there considerations to reach out to other national leaders?

3. Methods: During round two of the consensus, were participants given the list of interventions with or without the justifications and references? How many participants were involved in round 4? Did final consensus during round 4 require agreement from all present participants? A definition of consensus could be included. The wording of the 9-point Likert scale could also be included for methodological clarity.

4. Results: The authors should consider listing the interventions that were excluded from the final list as this can provide equally informative data for readers. Furthermore, it would be interesting to receive the associated explanations as to their lack of feasibility and/or effectiveness.

5. Results: The authors can consider including the following source about the development of postpartum hemorrhage bundles, which supports several perioperative interventions: Althabe F, Therrien MNS, Pingray V, Hermida J, Gülmezoglu AM, Armbruster D, et al. Postpartum hemorrhage care bundles to improve adherence to guidelines: A WHO technical consultation. Int J Gynecol Obstet. 2020;148(3):290–9.

6. Results: More background information regarding the current situation of availability of 24h emergency caesarean section services should be included.

Minor comments

- The logical flow on page 16, lines 319-325 is unclear and could be improved

Reviewer #2: A well written paper, and very valuable study on a very important clinical issue/challenge in Global Health.

Delphi study was appropriate to obtain expert consensus.

Perhaps more commentary on implementing policies to support the interventions outlined would enhance the discussion.

Reviewer #3: This is a well conducted and presented paper addressing a common cause of maternal mortalty. Care bundles are commonly referred to and used for obstetric hemorrhage (see Joint Council) and this paper addresses the peri-operative care bundle, particularly with respect to haemorrhage. (The one by the Joint Council may be referenced, even though it is more US based.)

The paper correctly included upstream (public health and proximate areas) that can be part of the care bundle.

All four rounds and the iterative process have been explained (Rounds 1,2, and 3 and finally the Round 4 discussion for remaining consensus). Noted that started with forty-four experts in round 1, which dropped to 41 and 40 in rounds 2 and 3 respectively. Did all forty-four rejoin in round 4?

Of the 14 potential interventions considered to have moderate effectiveness (median 4-6) and/or moderate feasibility (median 4-6), for consideration in the round four consensus meeting, four borderline interventions were upgraded to the final list of recommended interventions.

The reader can see the remaining ten interventions that did not make it to the final list. This is very useful for different countries to weigh the merits and demerits of those that were not included. This author noted a few not included that are constantly addressed in his setting as issues of concern based on regular maternal mortality reviews.

Strength – included obstetricians and anesthetists. The affiliations of the experts is noted. Were the specialists from different types of hospitals? For example, large teaching hospitals down to district level hospitals? Might this have made a difference?

The Limitations correctly mention that .. ‘no patients, midwives or public health specialists were involved in the Delphi process’. Agree that the involvement of these stakeholders may have widened the scope of recommendations. ‘Patient experience’ is becoming a useful tool to understand issues around quality of care. In the circumstance of peri-operative care, the views of theatre nurses and particularly recovery-ward nurses, if included, could have added more information. Nevertheless, a few ‘direct items’ like those related to early and increased surveillance and consideration of the monitoring for the shock index are what they may have brought up.

Thank you

6. PLOS authors have the option to publish the peer review history of their article (what does this mean?). If published, this will include your full peer review and any attached files.

**Do you want your identity to be public for this peer review?** For information about this choice, including consent withdrawal, please see our Privacy Policy.

Reviewer #1: **Yes: **Jenny Hoang Nguyen

Reviewer #2: No

Reviewer #3: No

---

## [Decision Letter · Decision Letter 1]

11 May 2022

Identifying interventions to reduce peripartum haemorrhage associated with caesarean delivery in Africa: a Delphi consensus study

PGPH-D-22-00210R1

Dear Dr. Biccard,

We are pleased to inform you that your manuscript 'Identifying interventions to reduce peripartum haemorrhage associated with caesarean delivery in Africa: a Delphi consensus study' has been provisionally accepted for publication in PLOS Global Public Health.

Best regards,

M. Dylan Bould

Academic Editor

Thanks for responding well to what were only fairly minor matters to consider - this paper is a fine contribution to the journal.

Reviewer Comments (if any, and for reference):

Reviewer's Responses to Questions

**Comments to the Author**

1. If the authors have adequately addressed your comments raised in a previous round of review and you feel that this manuscript is now acceptable for publication, you may indicate that here to bypass the “Comments to the Author” section, enter your conflict of interest statement in the “Confidential to Editor” section, and submit your "Accept" recommendation.

Reviewer #2: All comments have been addressed

Reviewer #3: All comments have been addressed

2. Does this manuscript meet PLOS Global Public Health’s publication criteria? Is the manuscript technically sound, and do the data support the conclusions? The manuscript must describe methodologically and ethically rigorous research with conclusions that are appropriately drawn based on the data presented.

Reviewer #2: Yes

Reviewer #3: Yes

3. Has the statistical analysis been performed appropriately and rigorously?

Reviewer #2: Yes

Reviewer #3: Yes

4. Have the authors made all data underlying the findings in their manuscript fully available (please refer to the Data Availability Statement at the start of the manuscript PDF file)?

Reviewer #2: Yes

Reviewer #3: Yes

5. Is the manuscript presented in an intelligible fashion and written in standard English?

Reviewer #2: Yes

Reviewer #3: Yes

6. Review Comments to the Author

Reviewer #2: No further comments, think all comments have been addressed

Reviewer #3: The responses to the review as well as those of the other two reviewers have been noted.

Thank you

7. PLOS authors have the option to publish the peer review history of their article (what does this mean?). If published, this will include your full peer review and any attached files.

**Do you want your identity to be public for this peer review?** For information about this choice, including consent withdrawal, please see our Privacy Policy.

Reviewer #2: No

Reviewer #3: No
